# *Sphingopyxis* sp. Strain OPL5, an Isoprene-Degrading Bacterium from the Sphingomonadaceae Family Isolated from Oil Palm Leaves

**DOI:** 10.3390/microorganisms8101557

**Published:** 2020-10-10

**Authors:** Nasmille L. Larke-Mejía, Ornella Carrión, Andrew T. Crombie, Terry J. McGenity, J. Colin Murrell

**Affiliations:** 1School of Environmental Sciences, University of East Anglia, Norwich NR4 7TJ, UK; n.mejia@uea.ac.uk (N.L.L.-M.); o.carrion-fonseca@uea.ac.uk (O.C.); 2School of Biological Sciences, University of East Anglia, Norwich NR4 7TJ, UK; a.crombie@uea.ac.uk; 3School of Life Sciences, University of Essex, Colchester CO4 3SQ, UK; tjmcgen@essex.ac.uk

**Keywords:** isoprene degradation, targeted isolation, DNA-SIP, *isoA*, isoprene monooxygenase, *Sphingopyxis*

## Abstract

The volatile secondary metabolite, isoprene, is released by trees to the atmosphere in enormous quantities, where it has important effects on air quality and climate. Oil palm trees, one of the highest isoprene emitters, are increasingly dominating agroforestry over large areas of Asia, with associated uncertainties over their effects on climate. Microbes capable of using isoprene as a source of carbon for growth have been identified in soils and in the tree phyllosphere, and most are members of the Actinobacteria. Here, we used DNA stable isotope probing to identify the isoprene-degrading bacteria associated with oil palm leaves and inhabiting the surrounding soil. Among the most abundant isoprene degraders of the leaf-associated community were members of the Sphingomonadales, although no representatives of this order were previously known to degrade isoprene. Informed by these data, we obtained representatives of the most abundant isoprene degraders in enrichments, including *Sphingopyxis* strain OPL5 (Sphingomonadales), able to grow on isoprene as the sole source of carbon and energy. Sequencing of the genome of strain OPL5, as well as a novel *Gordonia* strain, confirmed their pathways of isoprene degradation and broadened our knowledge of the genetic and taxonomic diversity of this important bacterial trait.

## 1. Introduction

Isoprene (2-methyl-1,3-butadiene) is a climate-active gas, with global emissions estimated at around 400–600 Tg per year, making it the most abundant biological volatile organic compound (BVOC) produced in the biosphere [1,2]. Unlike the potent greenhouse gas methane, which is produced in similar amounts, isoprene is a highly reactive compound that has multiple effects on climate and air quality [3]. In polluted urban regions where there are high concentrations of NOx, isoprene causes an increase in tropospheric ozone, whereas in pristine environments isoprene reacts with ozone, resulting in its depletion, and also with hydroxyl radicals which thus indirectly increases the residence time of methane in the atmosphere, leading to global warming [4]. Oxidation products of isoprene in the atmosphere form secondary aerosols that can increase cloud albedo and thus potentially result in global cooling [5]. Thus, isoprene is a climate-active BVOC with the potential to influence climate in several ways. Isoprene is produced by some bacteria, archaea, algae, fungi, protists, and animals, but the vast majority (>90%) of isoprene is from trees [6]. Not all trees produce isoprene, and the exact reasons why some trees can lose 1–2% of their photosynthetically fixed carbon as isoprene are not fully understood [7,8]. Isoprene has been shown to protect plants from heat and oxidative stress and has been implicated as a signaling molecule and regulator of expression of plant genes (reviewed in [9]). Isoprene-emitting trees include poplar and willow, which are often cultivated as biofuel crops, and oil palm, which is one of the highest isoprene-emitting trees [10,11]. Oil palm is a major crop plant grown in vast amounts particularly in southeast Asia, and is the source of around a third of the world’s vegetable oil [12]. Expansion of growth of such crop plants that produce isoprene has raised concerns about their long-term effects on air quality [13].

It has been known for a number of years that soils can be a sink for isoprene [14,15] and this has stimulated research on the microbiology of isoprene metabolism. The biological consumption of isoprene by bacteria and their role in the isoprene cycle has been reviewed in detail recently ([16,17,18]). In brief, a number of Gram-positive bacteria, mainly Actinobacteria of the genera *Rhodococcus* [19,20], *Gordonia*, *Mycobacterium* [21], and *Nocardioides* [22] have been isolated and characterized. The best studied is *Rhodococcus* strain AD45, isolated from freshwater sediment, which has become a model bacterium for the study of isoprene metabolism [23,24]. There are fewer cultivated Gram-negative bacteria that have been reported to degrade isoprene [25,26]. The most well-characterized of these is *Variovorax* strain WS11, originally isolated from soil [25,26,27]. All isoprene degraders characterized to date contain a multi-subunit isoprene monooxygenase, encoded by the genes *isoABCDEF*, which catalyzes the oxidation of isoprene to epoxyisoprene, and the gene cluster *isoGHIJ* encoding a glutathione transferase and enzymes catalyzing further steps in the isoprene oxidation pathway [23]. The isoprene monooxygenase (IsoMO) can be distinguished from related soluble di-iron monooxygenases (SDIMOs, [28]) that are involved in metabolism of alkenes, as well as alkanes and aromatic compounds (reviewed in [18]). In *Variovorax* strain WS11 and *Rhodococcus* strain AD45, *iso* genes are carried on megaplasmids [24,26]. New understanding of the physiology, biochemistry, and molecular biology of isoprene metabolism has provided the tools for designing functional gene primers and probes based on *isoA,* encoding a key component of the IsoMO [19,29]. Surveys with samples from the terrestrial environment, mainly soils and the phyllosphere have revealed the presence of isoprene-degrading bacteria, especially in environments containing trees that produce high concentrations of isoprene [19,25,27,29,30]. DNA-stable isotope probing (DNA-SIP) experiments have also revealed the diversity of active isoprene degraders present in these environments, which are often Actinobacteria, especially *Rhodococcus* species, but also include other Gram-negative isoprene degraders [19,25,27]. Oil palms may provide “hotspots” of isoprene production where isoprene degraders are likely to thrive, so we examined an oil palm tree in the Palm House at Kew Gardens, London. The aim of this study was to investigate the diversity of active isoprene-degrading bacteria in the phyllosphere and soil around an oil palm tree using DNA-SIP experiments. Here we report on these experiments, together with the isolation, characterization, and genome sequence of a new isoprene-degrading *Sphingopyxis* species.

## 2. Materials and Methods 

### 2.1. Sampling, Enrichment Assays, and DNA-SIP Experiments

Samples were collected (November 2016 and February 2017) from a 5-meter-high oil palm tree (germinated in 2002 and located in the Palm House, Kew Gardens, London, UK) and transported back to the laboratory for processing. During the first visit, leaflets and leaf swabs (three leaflets or approx. 34 g of leaves) were collected by cutting off a compound leaf from the oil palm tree. Swabs were collected on site according to Hedin et al. 2010 [31]. Leaves were also sampled for microbes by pressing them onto the surface of fresh, sterile Ewers minimal medium [32] (hereafter referred to as Ewers minimal medium) agar plates for preliminary enrichment and isolation of isoprene-degrading bacteria. In the laboratory, cells were detached from leaflets and leaf swabs by washing into Ewers minimal medium as previously described [27,31] and the resulting medium containing epiphytic cells was enriched with 20 ppmv of isoprene according to Crombie et al. 2018 [27] (Appendix A). Plates, inoculated directly by pressing leaves onto minimal medium agar plates, were incubated in sealed jars at 30 °C in the presence of approximately 1% (*v/v*) isoprene and checked for growth every day. During the second sampling, leaflets from the same oil palm tree and soil surrounding the base of the tree (at a depth of 0–5 cm) were taken for DNA-SIP experiments. DNA-SIP incubations were set up in triplicate using either 4 g soil in 40 mL of sterile water (as detailed in Larke-Mejía et al. 2019 [25]) or leaf washings with 40 mL of minimal medium (as described in [27]) in 2 liter sealed glass bottles supplied with 25–50 ppmv of either ^12^C- or ^13^C-labelled isoprene, as reported in [19], in order to promote enrichment of diverse isoprene degraders as seen in [25]. Isoprene consumption (Appendix A) in both sets of DNA-SIP experiments was closely monitored using gas chromatography as described previously [27]. ^13^C incorporation into biomass over time was then estimated (Appendix A). Isoprene was replenished frequently to restore the concentration of isoprene to between 25 and 50 ppmv throughout the DNA-SIP incubations [25]. Aliquots (approx. 10 mL) of the DNA-SIP incubations were harvested before enrichment (T_0_) and after incorporation of approximately 25, 50, and 75 µmol g^−1^ of ^13^C (Appendix A). Aliquots were stored at −20 °C until DNA extraction with FastSpin DNA soil kit (MP Biomedicals, Santa Ana, CA, US) following the manufacturer’s instructions.

DNA (soil: 5 µg, leaves: up to 1 µg) was separated into ^13^C-labelled (“heavy”) DNA and ^12^C-unlabelled (“light”) DNA fractions by isopycnic ultracentrifugation as described previously [19]. DNA concentration and density of each fraction was determined with a Qubit^TM^ dsDNA HS Assay kit (Thermo Fisher Scientific, UK) and an AR200 digital refractometer (Reichert, USA), respectively. Denaturing gradient gel electrophoresis of 16S rRNA gene amplicons (not shown) and fractionation plots of DNA abundance against fraction density (Appendix A) for each time-point were used to determine which fractions contained “heavy” and “light” DNA to be used for further analysis. Comparing fractionation plots from different time-points in both experiments also confirmed the progressive incorporation of the ^13^C-label during the enrichment, observed as the migration of DNA to “heavy” and “light” regions of ultracentrifuge tubes in ^13^C incubations compared to only a “light” region for ^12^C incubations [27], and development of a larger “heavy” peak as the isoprene enrichment increased.

### 2.2. Identification of Active Isoprene-Degrading Bacteria from Oil Palm Soils and Leaves 

The datasets generated for this study can be found in the GeneBank repository (https://www.ncbi.nlm.nih.gov/bioproject/PRJNA272922). Bacterial community profiles from unenriched (T_0_) and ^13^C-enriched DNA at each time-point were analyzed by partial 16S rRNA gene sequencing (performed with 341f and 785r primers, [33]) utilizing an Illumina MiSeq platform and carried out at MR DNA (Molecular Research LP) (Shallowater, TX, USA). 16S rRNA gene sequences (SRA accession number SRR12533424) from these PCR amplicons were processed as described by Larke-Mejía et al. (SRA accession number SRR12533424) [25].

### 2.3. Enrichment and Isolation of Isoprene-Degrading Bacteria from Oil Palm Samples

At the end of DNA-SIP experiments, 10 mL samples were transferred to 120-mL glass vials containing 10 mL of fresh Ewers minimal medium [32] pH 6.5, supplemented with 1 ml per liter vitamins v10 solution (DSMZ). Vials were then sealed with butyl rubber stoppers and isoprene was injected to maintain a concentration of 25–50 ppmv. After several subcultures, isoprene-degrading bacteria were selected and purified as previously described [25]. For subsequent characterization, isoprene-degrading strains were grown in Ewers minimal medium containing 1% (*v/v*) isoprene. DNA from cultures was extracted using the FastSpin DNA soil kit (MP Biomedicals, Santa Ana, CA, USA) according to the manufacturer’s instructions, and used as a template for PCR amplification of 16S rRNA genes [34] and *isoA* genes (using primers *isoA*14f and *isoA*1019r, [29]). The 16S rRNA gene was used to identify isoprene-degrading isolates of interest by comparing with 16S rRNA gene sequences from characterized isoprene-degrading bacteria, and also by comparing with 16S rRNA genes from the most abundant OTUs during the DNA-SIP experiments (targeted isolation). 16S rRNA gene sequence alignments and maximum likelihood trees of 16S rRNA genes were constructed as described previously [25].

### 2.4. Genome Sequencing and Analysis

The genomes of the new isoprene-degrading bacteria *Sphingopyxis* strain OPL5 (GenBank accession CP060725.1) and *Gordonia* strain OPL2 (NCBI Reference Sequence NZ_RKME00000000.1) were sequenced and assembled by MicrobesNG (University of Birmingham, Birmingham, UK) using Oxford Nanopore and Ilumina HiSeq 2500 platforms, respectively. After growth on isoprene and confirmation of purity by microscopy, *Sphingopyxis* strain OPL5 and *Gordonia* strain OPL2 were grown at 30 °C for 7 days on minimal media plates supplemented with approximately 1% (*v/v*) isoprene supplied to the headspace in a sealed jar. The biomass was collected from plates and deposited to bead tubes supplied by MicrobesNG (University of Birmingham, UK). Ilumina genome sequencing was conducted by MicrobesNG as follows: “Three beads were washed with extraction buffer containing lysozyme and RNase A and incubated for 25 min at 37 °C. Proteinase K (0.05 µg/mL) and RnaseA (0.1 µg/mL) were added and incubated for 5 min at 65 °C. Genomic DNA was purified using an equal volume of SPRI beads and resuspended in EB buffer. DNA was quantified in triplicate with the Quantit dsDNA HS assay in an Eppendorff AF2200 plate reader. Genomic DNA libraries were prepared using Nextera XT Library Prep Kit (Illumina, San Diego, CA, USA) following the manufacturer’s protocol with the following modifications: two nanograms of DNA instead of one were used as input, and PCR elongation time was increased to 1 min from 30 s. Pooled libraries were quantified using the Kapa Biosystems Library Quantification Kit for Illumina on a Roche light cycler 96 qPCR machine. Libraries were sequenced on the Illumina HiSeq instrument using a 250 bp paired end protocol. Reads were adapter trimmed using Trimmomatic version 0.30 (26) with a sliding window quality cutoff of Q15. *De novo* assembly was performed on samples using SPAdes version 3.7 [35].” For the enhanced genome sequencing, libraries using Oxford Nanopore were prepared as follows: “Broth cultures of each isolate were pelleted and the pellet was resuspended in the cryoperservative of a Microbank™ (Pro-Lab Diagnostics UK, United Kingdom) tube and stored in the tube. Approximately 2× 10^9^ cells were used for high molecular weight DNA extraction using Nanobind CCB Big DNA Kit (Circulomics, Baltimore, MD, USA). DNA was quantified with the Qubit dsDNA HS assay in a Qubit 3.0 ((Invitrogen) Eppendorf UK Ltd., Stevenage, UK). Long read genomic DNA libraries were prepared with an Oxford Nanopore SQK-RBK004 kit and/or SQK-LSK109 kit with Native Barcoding EXP-NBD104/114 (ONT, Oxford, UK) using 400–500 ng of HMW DNA. Twelve to twenty-four barcoded samples were pooled together into a single sequencing library and loaded in a FLO-MIN106 (R.9.4 or R.9.4.1) flow cell in a GridION (ONT, UK). The contigs were annotated using Prokka 1.11. [36]”

### 2.5. Genome Analysis

Genome characteristics were assessed with the use of the Rapid Annotation using Subsystem Technology (RAST server) version 2.0 at https://rast.nmpdr.org/ [37] and the MicroScope Microbial Genome Annotation and Analysis Platform version 3.13.5 at https://mage.genoscope.cns.fr/microscope [38]. These two platforms were used to determine the general characteristics of the genomes of *Sphingopyxis* strain OPL5 and *Gordonia* strain OPL2 and to search for the genes of interest. Amino acid sequences that were likely candidates for enzymes involved in the isoprene degradation pathway were compared to a personally curated database of such proteins with the use of BLASTp https://blast.ncbi.nlm.nih.gov/Blast.cgi [39]. The Microbial Genome Atlas (MiGA) http://microbial-genomes.org [40] was used to determine taxonomic affiliation, novelty and gene diversity with the use of the NCBI prokaryotic genome database. Gene Graphics [41] was used to visualize the isoprene metabolic genes for Figure 3.

### 2.6. Isoprene Oxidation Assays

*Sphingopyxis* strain OPL5 was grown in a 4 L working volume fermenter (Electrolab, Tewkesbury, UK) with Ewers minimal medium and isoprene as the sole carbon and energy source. Optimal growth conditions for *Sphingopyxis* strain OPL5 were used (30 °C, 160 rpm, pH 6.5, 1 mL per liter vitamins v10 solution, and air flow 2.4 L/min). Isoprene was supplied by bubbling air (1 mL min^−1^) through a small volume of liquid isoprene contained in a 30-mL vial, held on ice (as described in [26]). Cells were harvested three times, at an optical density (540 nm) of 1.0, 2.5, and 3.1, respectively (Appendix A) by centrifuging at 7000× *g* for 20 min and resuspending in 50 mM HEPES, pH 6.5, before snap-freezing and storing at −80 °C. Frozen cells retained full isoprene oxidation activity over several months. Isoprene oxidation rates were calculated using a Clark-type oxygen electrode [42]. Frozen cells were thawed and resuspended to an OD_540_ of 2.0 in 3 mL of 0.5 M phosphate buffer (pH 6.5) and equilibrated to 30 °C for 2–3 min with stirring. Substrate-induced rates of oxygen depletion were calculated for concentrations of isoprene from 3 μM to 30 μM (Appendix A) as detailed in [26].

## 3. Results and Discussion

### 3.1. Testing Methods to Recover Leaf Epiphytes

In preparation for DNA-SIP incubations, an initial experiment was carried out to determine the better method to recover epiphytes from the leaves. Epiphytes were recovered either by washing cells from leaves or by removing with swabs using the methods described by Hedin et al. 2010 [31]. Cells were then incubated with 25 ppmv isoprene and the isoprene consumption rate with cells retrieved by oil palm leaf (OPL) washings and swabbing (OPL swabs) was monitored (Appendix A) and a lag in initiation of isoprene consumption was observed with both incubations. After isoprene consumption commenced (approx. 110 h) the rate of isoprene consumption with bacterial cells retrieved by leaf-washing was marginally faster than with leaf swabs, and so the leaf-washing method was used subsequently. 

### 3.2. Active Isoprene-Degrading Bacteria Associated with an Oil Palm Tree

In order to determine the diversity of isoprene degraders associated with an oil palm tree, two separate DNA-SIP experiments were performed in triplicate using: (a) Cells washed from the surface of oil palm leaves (hereafter termed leaf-washings), to recover epiphytic cells present on leaves; and (b) soil from the vicinity of this tree. ^13^C-labelled isoprene (with ^12^C-isoprene controls) was added to 25 ppmv and the consumption of isoprene for each microcosm was monitored over approximately 6–9 days. Over the time-course of the DNA-SIP experiments, between approximately 25 and 86 µmol ^13^C was incorporated per gram of soil or per gram of washed leaf which is within the range of ^13^C incorporation required for a successful DNA-SIP experiment, as recommended by Neufeld et al. [43]. DNA-SIP incubations with soil began consuming isoprene after approximately 2 days of incubation, while incubations with leaf washings consumed isoprene after a lag of approximately 4 days. Once isoprene was depleted, it was added again to 25 ppmv over the course of the DNA-SIP experiments, and the repeated supply of isoprene after its depletion led to rapid degradation (Appendix A). The total amount of isoprene consumed by the incubations was used as a proxy for the amount of ^13^C-label incorporation into biomass and this allowed the selection of harvesting times for samples from DNA-SIP microcosms for DNA extraction at three time points corresponding to 25, 50, and 75 µmol of ^13^C-label incorporated per gram of starting material (142, 166, and 190 h for soils and approximately 220, 280, and 310 h for leaf washings, Appendix A and Appendix A). DNA-SIP enrichments were harvested at these times based on the assumption that approximately half of the carbon, added to enrichments as ^13^C-isoprene, was incorporated into biomass, and estimates of the amount of ^13^C-label incorporated are summarized in Appendix A. This yielded eight samples (each in triplicate) for soil and leaf washing incubations, at time points T_1_, T_2_, and T_3_, with T_0_ being a sample retained at the start of DNA-SIP incubations.

DNA was extracted from samples and then fractionated to separate heavy ^13^C-DNA from light ^12^C-DNA. Comparison of ^13^C-DNA and ^12^C-DNA profiles (controls) across ^13^C-isoprene and ^12^C-isoprene SIP enrichments confirmed the incorporation of ^13^C-label into microbial biomass over the course of the incubations with both soil samples and leaf-washings (Appendix A). The fractions representing ^12^C-DNA (light) and ^13^C-DNA (heavy) samples obtained from soil and leaf washings, taken at T_1_, T_2_, and T_3_, are also indicated in Appendix A. The change in the ratio of heavy DNA to light DNA throughout the DNA-SIP experiments was different between soils and leaves because soils contained higher biomass at the beginning of the enrichment, shown as a larger ^12^C-unlabeled peak.

### 3.3. Bacterial Community Composition in SIP Enrichments

Changes in the bacterial communities in DNA-SIP enrichments were followed by comparing the labelled 16S rRNA gene profiles in samples that were incubated with ^13^C-isoprene at T_1_, T_2_, and T_3,_ with the T_0_ community. Appendix A and Figure 1 depict the relative abundance of 16S rRNA genes present in these samples, at the order and genus level respectively, and the changes in bacterial community composition over the time-course of the DNA-SIP experiments. For clarity, in Appendix A and Figure 1, we followed the enrichment of replicate 1 throughout enrichment (T_1_ to T_3_, shown as T1.1, T2.1 and T3.1) and the T_3_ samples are depicted as T3.1, T3.2, and T3.3 to show the variation in relative abundance of 16S rRNA genes across replicates at the end of the enrichment experiments. 16S rRNA gene analysis of non-enriched soil (T_0_) and non-enriched leaves (T_0_) revealed the diversity of organisms from six main orders (Appendix A): the Rhizobiales (13.4% in soil and 13.9% in leaves), the Actinomycetales (9.3% and 13.3%, respectively), Bacillales (2.8% and 8.7%, respectively), Sphingomonadales (1.5% and 7.7%, respectively), Sphingobacteriales (2.2% and 7.6%, respectively), and Burkholderiales (2.9% and 4.6%, respectively). The order Acidobacteriales was also abundant only in soils (8.9%). After enrichment with ^13^C-labelled isoprene, ^13^C labelled heavy DNA samples from both soils and leaves were dominated by Actinomycetales (21–60% and 7.6–69.3%, respectively) and Burkholderiales (8.9–68.6% and 3.7–38.6%, respectively). Of significant interest was the substantial enrichment of 16S rRNA genes from the Sphingomonadales, at 7–57% relative abundance in ^13^C-DNA samples from DNA-SIP experiments with oil palm leaf-washings (Appendix A). 

### 3.4. ^13^C—Labelling of Putative Isoprene Degraders in Leaf Washings

Analysis at the genus level (Figure 1) revealed the presence in leaf washing incubations of labelled members of the genus *Gordonia* in leaf washing incubations, (4.4% at T_1_ up to 54.2% at T_3.2_) some of which are known to grow on isoprene [21]. Sequences affiliated with *Aquincola*, a member of the Comamonadaceae family (along with the isoprene degraders *Variovorax* strain WS11 and *Ramlibacter* strain WS9 [25,26,27]) were also present in ^13^C-DNA. Approximately 22–31% relative abundance of *Aquincola*-like sequences in T_1_ and T_2_ samples and one T_3_ sample (T_3.1_) from leaf washing SIP incubations, but were less abundant in two of the T_3_ samples (T_3.2_ and T_3_._3_) suggesting that this might have been an as-yet unidentified isoprene-degrader that was outcompeted by other isoprene degraders over time. An example of an important member of this genus, found to degrade important alkenes, is *Aquincola tertiaricarbonis* L108. This strain is an example of a member of this genus able to degrade aerobic fuel oxygenate, *tert*-alkyl ethers, and alcohols by expression of cytochrome P450, EthABCD monooxygenase, and possibly the tertiary alcohol monooxygenase MdpJ which produces and then degrades the hemiterpene 2-methyl-3-buten-2-ol [44,45].

Of significance was the presence of relatively high numbers of 16S rRNA gene sequences from *Sphingomonas* in oil palm leaf washings (6.9–56.3% relative abundance), especially dominating ^13^C-DNA samples from T_3.1_. Thus far there were no known isoprene-degrading *Sphingomonas* strains and so this became a target for isolation (see later). Also of note was the enrichment of members of the genus *Methylobacterium* in DNA-SIP experiments with oil palm leaf washings (Figure 1) with the relative abundance of 16S rRNA genes in ^13^C-DNA increasing from 3.4% at T_0_ to 9.6% at T_2_ and to approximately 2.7–10.6% relative abundance in T_3_ samples. This was interesting because *Methylobacterium* is well-known as an inhabitant of the phyllosphere since it grows on methanol produced from the degradation of pectin [46] and it has also been reported to be an isoprene-degrader [47]. Despite our efforts, we were not subsequently able to isolate an isoprene-degrading *Methylobacterium* but recently McGenity and colleagues ([48] and manuscript in preparation) have reported the isolation of isoprene-degrading *Methylobacterium* from poplar and willow leaves.

### 3.5. ^13^C—Labelling of Putative Isoprene Degraders in Soil

In ^13^C-DNA samples retrieved from DNA-SIP enrichments using soil from the vicinity of the oil palm tree, 16S rRNA genes from the genera *Rhodococcus* (19–55% relative abundance) and *Aquabacterium* (from the Comamonadaceae family, 7–64% relative abundance) dominated the ^13^C-isoprene enrichments across all time points (Figure 1). *Rhodococcus* strain AD45 is a well-characterized isoprene degrader that has become a model for the study of isoprene metabolism [24]. *Rhodococcus* has also been observed in high abundance in other DNA-SIP experiments with estuarine samples [21], willow soils [19,25], poplar leaves [27] and so it was perhaps not surprising to see members of this genus enriched in oil palm soil samples. Analysis of ^13^C-DNA samples from SIP enrichments with soil also revealed the enrichment of *Saccharibacter* from the Rhodospirillales order (present at a relative 16S rRNA gene abundance of around 3–18%) which was also interesting because there are also no reports of isoprene-degrading isolates from this genus. Labelled *Gordonia* 16S rRNA gene sequences were also noted from soil for all enriched time-points (present at around 2–4% relative abundance) which was not unexpected as isoprene-degrading *Gordonia* have previously been reported [21].

### 3.6. Targeted Isolation of Isoprene Degraders from Oil Palm

For the targeted isolation of isoprene degraders identified by the 16S rRNA gene profiling of bacterial communities in DNA extracted from SIP experiments (Figure 1), the DNA-SIP microcosms were continuously enriched with 25 ppmv of isoprene and samples were sub-cultured into Ewers minimal medium supplied with 25 ppmv isoprene. Subsequent serial dilution, plating of enrichments, and *isoA* PCR were used to confirm the presence of the *isoA* gene of *Rhodococcus* sp. strain OPL1, *Gordonia* strain OPL2, and *Sphingopyxis* strain OPL5 from oil palm leaf enrichments. From the strains isolated, the latter two were of most interest because members of these families were found to be enriched during DNA-SIP experiments with leaf washings from oil palm (Figure 1). Most importantly, there is only one other isolated isoprene-degrading representative from the *Gordonia* genus (*Gordonia* i37, [21]) and no *Sphingopyxis* have been reported previously to grow on isoprene. Unfortunately, *Aquabacterium*, *Saccharibacter,* and *Methylobacterium* from which 16S rRNA gene sequences of which were enriched in ^13^C-DNA from the SIP experiments described above, were not isolated. All three isolates that were obtained subsequently grew well (OD_540_ > 1.0) on Ewers minimal medium containing up to 1% (*v/v*) isoprene as their sole carbon and energy source. Since *Rhodococcus* species growing on isoprene have previously been described in detail [19,20,24], *Rhodococcus* strain OPL1, which has 100% 16S rRNA gene sequence identity to *Rhodococcus* strain MAK1 from a coal tar-contaminated aquifer located in South Glens Falls, New York [49], was not characterized further.

The pink-pigmented oil palm leaf isolate *Gordonia* strain OPL2 (growth rate 0.12 h^−1^) was the second isoprene degrader isolated from this genus and the first isoprene degrader from the *Gordonia* genus isolated from the phyllosphere of an isoprene-producing tree. It had 97.53% sequence identity with the 16S rRNA gene of *Gordonia polyisoprenivorans* strain i37, an isoprene degrader isolated from an estuary [21]. The closest relative of strain OPL2 in terms of 16S rRNA gene was *Gordonia* strain 647 W.R.1a.05, at 99.92% sequence identity, isolated from the venom duct of the cone snail, *Conus circumcises* [50] and *Gordonia terrae* strain DSM 43249, with 99.55% sequence identity (Appendix A).

*Sphingopyxis* strain OPL5, a member of the Sphingomonadaceae in the Alphaproteobacteria, was isolated from isoprene enrichments arising from the oil palm leaf-washing DNA-SIP experiments. This yellow-pigmented isolate had high 16S rRNA gene sequence identity with the characterized *Sphingopyxis panaciterrae* strain Gsoil124 (99.48% 16S rRNA gene sequence), which was isolated from a soil sample taken from a ginseng field in Pocheon Province, South Korea [51] (Appendix A). In a survey of *isoA* gene diversity, Carrión et al. (2018) [29] reported the presence of *Sphingopyxis*-like *isoA* from the same oil palm leaves described here (60% relative abundance of *Sphingopyxis*-like *isoA* sequences), and from freshwater sediment (>40%) and poplar leaves. *Gordonia*-like *isoA* sequences were found in poplar leaf samples, in oil palm leaves from this study (40% of sequences affiliated with *Gordonia*) and Malaysian oil palm leaves (over 98%) [29]. In a similar DNA-SIP experiment, enriching leaves and soil from a Malaysian oil palm tree, Carrión et al. showed the presence of *Gordonia* and members of the Sphingomonadaceae family enriched in the soil and phyllosphere communities [30].

In order to assess the relative abundance and importance of these isolates during the enrichments, two phylogenetic trees were constructed, placing the abundant OTUs obtained from oil palm leaf SIP in context with the isolates obtained here together with representatives from the Sphingomonadaceae family (Figure 2A) and representatives of the *Gordonia* genus (Figure 2B). The most abundant members of the Sphingomonadaceae were *Sphingomonas*-like OTU 240 and OTU 2 which cluster closely with *Sphingomonas polyaromaticivorans* B2-7 and *Sphingomonas oligoaromativorans* strain IMER-A1-28, and OTU 37 (18.7%, 7.5%, and 1.2% relative abundance at T_1_, respectively). The most abundant *Gordonia*-like sequences were, OTU 4 (2.5% at T_1_, 36.4% at T_2_, and 39% at T_3_), identical to estuarine isolate *Gordonia polyisoprenivorans* strain i37 [21], OTU 5 (0.6% at T_1_, 2.9% at T_2_, and 1.8% at T_3_), identical to our isolate *Gordonia* OPL2, and OTU 571 (up to 3.5% abundant at T_2_), distinct from known isoprene-degrading isolates (Figure 2B). According to the DNA-SIP amplicon sequencing data, *Gordonia* strain OPL2 was the second most abundant isoprene-degrader from the *Gordonia* genus in the enrichments. Although *Sphingopyxis* strain OPL5 was not among the abundant isoprene degraders during DNA-SIP enrichments with oil palm leaf washings (Figure 1), targeted isolation, which was informed by these SIP experiments, enabled the isolation of Sphingopyxis strain OPL5 as a representative isoprene-degrader of the Sphingomonadaceae, as well as *Gordonia* strain OPL2.

### 3.7. Genome Sequencing and Characterization of Isoprene Gene Clusters

The genomes of strains OPL2 and OPL5 were sequenced to compare the newly isolated *Gordonia* strain OPL2 with *Gordonia* strain i37 [21] and to provide deeper insights into the mechanisms of isoprene metabolism in *Sphingopyxis* strain OPL5, the first strain of this genus found to grow on isoprene. The *Gordonia* strain OPL2 genome was 5.8 Mbp and had a GC content of 67.3% GC which is very similar to that of *Gordonia* i37 at 6.2 Mbp and a GC content of 66.8% (Table 1). 

The complete genome of *Sphingopyxis* strain OPL5 comprised of one contig of 4.67 Mbp and had a GC content of 65.89% (Table 1). The genome size and GC content are comparable with those of other *Sphingopyxis* species, the genomes of which vary considerably within the broad range of 3.0–6.0 Mbp [52]. Specifically, the genome of the closest sequenced relative of strain OPL5, *Sphingopyxis* QXT-31 (see Figure 2), was 4.3 Mbp and had a GC content of 66.5%. Focusing on genes that are known to be involved in isoprene metabolism [24], the organization of *iso* genes from *Gordonia* sp. strain OPL2 and *Sphingopyxis* strain OPL5 were compared with those of the most well-characterized *iso* gene cluster of *Rhodococcus* strain AD45 [24], the estuarine isoprene-degrader *Gordonia* i37 [21] and the most well-characterized Gram-negative isoprene degrader *Variovorax* strain WS11 [25,26] (Figure 3). Figure 3 and Appendix A also include the comparison of the iso gene cluster of strain OPL5 with a partial *iso* cluster from a *Sphingopyxis-*like MAG (metagenome assembled genome, wsMG4), recovered from a DNA-SIP experiment performed with willow soil [25]. The recovery of this MAG suggests the presence of other isoprene-degrading bacteria from the Sphingomonadales order in the willow soil environment.

The cluster shows similarities and differences in the organization of the 10 main *iso* genes in isoprene degraders. The first difference observed is that all *iso* genes of the Gram-positive model isoprene degrader *Rhodococcus* strain AD45 are clustered together (*isoGHIJABCDEF*), while other isoprene clusters have a *aldH1* gene, coding for an aldehyde dehydrogenase, between *isoJ* and *isoA* (Figure 3); *Rhodococcus* strain AD45 has the gene *aldH1* upstream *isoG*. When comparing the protein sequence similarity, all *iso* gene products from Gram-positive isoprene degraders have closer percentage similarity between each other, and the same is observed when *iso* gene products from Gram-negative bacteria are compared. For example, IsoA is 86–87% similar between both *Gordonia* isolates and *Rhodococcus* strain AD45, while the number decreases to 72–73% when comparing the same IsoA sequence from *Rhodococcus* AD45 with IsoA from the Gram-negative isoprene-degraders *Variovorax* strain WS11 and *Sphingopyxis* strain OPL5.

### 3.8. Sphingopyxis Strain OPL5 Growth and Affinity for Isoprene 

*Sphingopyxis* strain OPL5 grew best at 30 °C, with shaking at 160 rpm, at a pH between 6.0 and 7.0, and while supplementing Ewers medium with 1 mL per liter v10 vitamins solution. *Sphingopyxis* strain OPL5 grown with the previous conditions and supplemented with 1% *v/v* isoprene had a specific growth rate of 0.14 h^−1^ and doubling time of approximately 5.0 h. Interestingly, the growth rate was very similar between isoprene concentrations of 1–10% (*v/v*), with no inhibition at higher concentrations of isoprene. A morphological change to elongated rods at the highest isoprene concentration was also noted. These optimized conditions were then used to grow *Sphingopyxis* strain OPL5 in a fed-batch bioreactor with Ewers minimal medium and isoprene as the sole carbon and energy source, resulting in the growth rate varying with a doubling time of around 3.6 to 8.4 hrs (Appendix A). 

In order to estimate substrate-induced oxidation rates, using concentrations of isoprene between 3 μM and 30 μM, *Sphingopyxis* strain OPL5 cells were used in oxygen-electrode experiments. Km and Vmax values were calculated using Lineweaver-Burk, Eadie-Hoffstee, and Hanes plots, yielding values of 2.6 μM and 10.6 nmol/min/mg; 2.2 μM and 10.1 nmol/min/mg; 2.8 μM and 10.7 nmol/min/mg dry weight of cells, respectively (Appendix A). The rates of oxidation of isoprene by *Sphingopyxis* strain OPL5 were approximately three times lower than *Variovorax* strain WS11 (31.2 nmol/min/mg [26]) and seven times lower than *Rhodococcus* strain AD45 (reported oxidation rates of 76 nmol min^−1^ mg^−1^ [53]).

## 4. Conclusions

DNA-SIP enrichment experiments with soil and leaves from an oil palm tree in the UK provided insights into the diversity and abundance of active isoprene-degrading bacteria that are present on and around this high isoprene-emitting species of tree. Informed by 16S rRNA gene amplicon sequencing, we carried out targeted isolation of new isoprene degraders in DNA-SIP enrichments. The isolation of three isoprene-degrading isolates from the phyllosphere of this oil palm tree, from taxonomic groups identified as active by DNA-SIP experiments, highlights the importance of using cultivation-independent methods to inform cultivation-dependent strategies. The genomes of *Gordonia* strain OPL2 and *Sphingopyxis* strain OPL5 have increased the known diversity of isoprene degraders and their *iso* genes from different environments. Isolation and characterization of *Sphingopyxis* strain OPL5 has also been important in enabling the identification of *Sphingopyxis*-like sequences recovered from a wide range of environments in previous cultivation-independent studies and provides further insights into the broader diversity of isoprene-degrading Proteobacteria from the Sphingomonadales family.

## Figures and Tables

**Figure 1 microorganisms-08-01557-f001:**
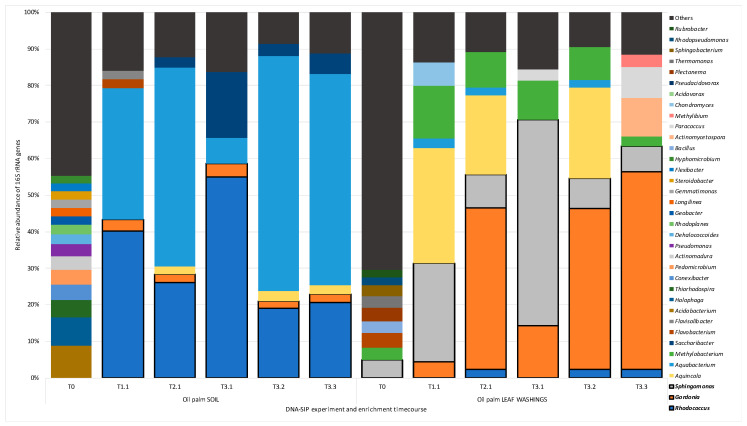
**Relative abundance of bacterial genera during DNA-stable isotope probing (DNA-SIP) enrichments**. Figure 1 shows the relative abundance of bacterial 16S rRNA genes (at the genus level), obtained by PCR of enriched and un-enriched DNA extracted from oil palm soil samples and leaf washings. Results include un-enriched (un-fractionated) samples and heavy DNA fractions (Appendix A) for three time points T_1_, T_2_ for one replicate (shown as T1.1, T2.1) and T_3_ for all three replicates (shown as T3.1, T3.2, and T3.3; refer to Appendix A for the time course). Only 16S rRNA gene sequences with a relative abundance of greater than 2% or over are shown. 16S rRNA gene sequences with a relative abundance of less than 2% are grouped together as “others.” Sequences from the genera (*Rhodococcus*, *Gordonia*, and *Sphingomonas*), highlighted with a black border, were identified as putative isoprene-degrading bacteria according to ^13^C-labelling.

**Figure 2 microorganisms-08-01557-f002:**
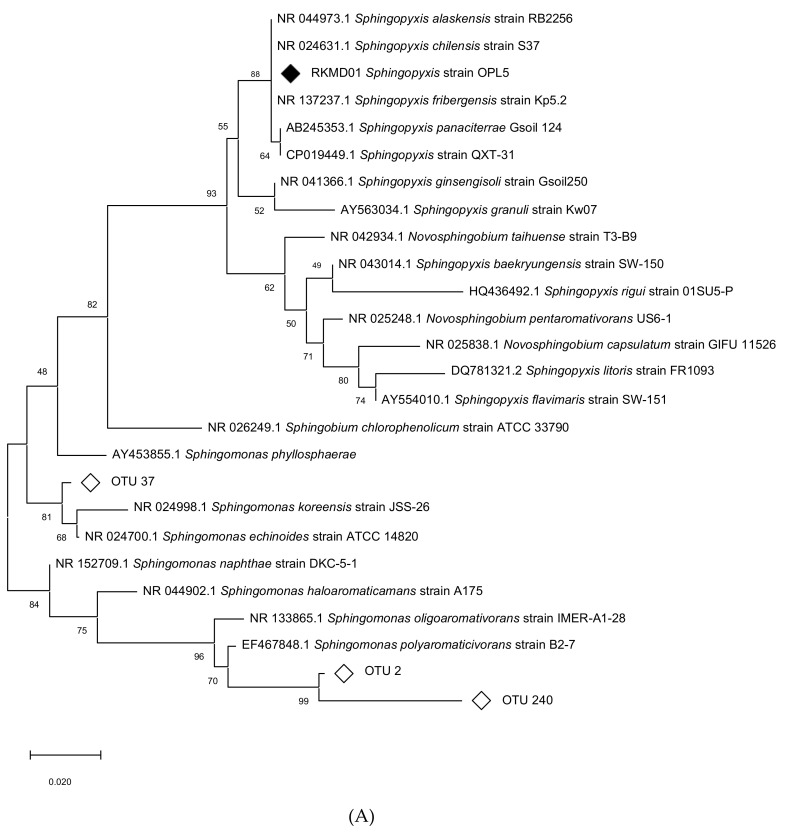
Partial 16S rRNA gene phylogenetic trees showing the oil palm leaf isolates, the most abundant closely related OTUs during DNA-SIP enrichments and related representatives from the databases: *Sphingopyxis* strain OPL5 (A) and *Gordonia* strain OPL2 (B).Trees were constructed using the Maximum Likelihood method using the partial 16S rRNA gene (V3 to V4 variable regions) amplified using 341f and 785r primers [1]. Following removal of gaps and missing data, there were 418 bp (A) and 409 bp (B) in the alignment. Bootstrap values (1000 replications) are shown. Strains isolated in this study are indicated with black diamonds and the most abundant closely related OTUs with empty diamonds. Relative abundance (%) of the most abundant OTUs before (T_0_) and throughout the DNA-SIP experiments (T_1_, T_2_, and T_3_) for soils and leaf washings were as follows: A) Sphingomonadaceae-like OTU 240 (in soils < 0.6% and on leaves 1.4% at T_0_, 18.7% at T_1_), OTU 2 (in soils <0.4% and leaves 7.5% at T_1_) and OTU 37 (in soils <0.1% and leaves 1.2% at T_1_). B) *Gordonia*-like: OTU 4 (in soils 1.5% at T_1_ and up to 39.0% at T_3_), OTU 571 (soils <0.2% and leaves up to 3.8% at T_2_) and OTU 5 (soils up to 1.1% at T_1_ and leaves 2.9% at T_2_). The scale bar shows nucleotide substitutions per site.

**Figure 3 microorganisms-08-01557-f003:**
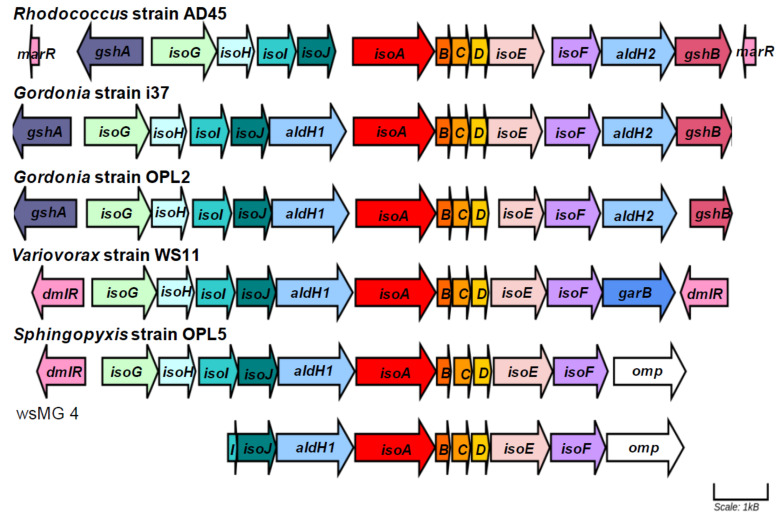
Isoprene metabolic gene clusters. Isoprene metabolic gene clusters in isoprene-degrading isolates (including *Sphingopyxis* strain OPL5 and *Gordonia* strain OPL2 isolated in this study), and a representative contig with the *iso* gene cluster obtained from metagenome co-assembly of ^13^C-DNA from isoprene-enriched willow soil (wsMG4). The *isoA* gene that codes for the alpha-subunit of the monooxygenase is shown in red. The % identity of *iso* gene-encoded polypeptides to the corresponding Iso polypeptides of the well-characterized *Rhodococcus* strain AD45 are shown in Appendix A*. omp*: outer membrane protein transfer.

**Table 1 microorganisms-08-01557-t001:** General characteristics of the genomes of *Gordonia* strain OPL2 and *Sphingopyxis* strain OPL5.

	*Gordonia* OPL2	*Sphingopyxis* OPL5
NCBI Tax ID	2486274	2486273
Length (bp)	5,759,526	4,676,975
GC (%)	67.3	65.89
Contigs	132	1
N50	80,039	4,676,975
CDS (total)	5313	4403
Genes (coding)	5200	4392
Genes (RNA)	55	51
rRNAs (5S, 16S, 23S)	3, 3, 1	1, 1, 1
tRNAs	45	55
Pseudogenes (total)	113	69
Coding Density (%)	91.2	91.9

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
