# Peer review of "Sphingopyxis sp. Strain OPL5, an Isoprene-Degrading Bacterium from the Sphingomonadaceae Family Isolated from Oil Palm Leaves"

_microorganisms, 2020, doi:10.3390/microorganisms8101557_

Round 1

Reviewer 1 Report

Larke-Mejia et al. describe the environmental assessment of the isoprene-degradation community from oil palm leaves. The authors performed the DNA-SIP survey of the microbial community and followed up with an enrichments study, strain isolations, and novel isoprene-degraders' genome sequencing. It is a substantial study, well written, and well presented. I have only a few minor comments:

  1. Targeted strain isolation (i.e., in Abstract- Informed by these data; in Results - Targeted isolation of isoprene degraders from oil palm, etc.). From the description provided, It was not clear how genetic community analyses directed the enrichment strategy. What were modifications to the enrichment strategies made based on the genetic information obtained from DNA-SIP? 
  2. In the abstract, the authors indicated that most tree phyllosphere capable of using isoprene are Actinobacteria members. Are there any significant experimental changes for this study to pick up mostly Proteobacteria from the phyllosphere? Maybe highlight this experimental knowhows in Materials and Methods.  
  3. Why were the enrichment and genomic strategies focused primarily on Sphingopyxis and Gordonia? Can the isolates be named " the most abundant isoprene degraders"? 
  4. Fig. 1 Consider changing the coloring patterns; it is hard to separate Methylibium from Pseudomonas and Aquincola. 

Author Response

  1. Targeted strain isolation (i.e., in Abstract- Informed by these data; in Results - Targeted isolation of isoprene degraders from oil palm, etc.). From the description provided, It was not clear how genetic community analyses directed the enrichment strategy. What were modifications to the enrichment strategies made based on the genetic information obtained from DNA-SIP? 

Targeted strain isolation refers to the use of 16S rRNA amplicon sequencing information to inform the isolation of isoprene-degraders of interest. The information obtained from Figure 1 informed us about families in Actinobacteria (Gordonia and Rhodococcus) and Proteobacteria (Aquincola, Aquabacterium and Sphingomonas) that we could try to isolate and the focus was on isolating more Gram negative strains since Actinobacterial isoprene degraders are now fairly well documented. As mentioned in the manuscript (lines 315-323), we were able to isolate Rhodococcus, Gordonia and Sphingopyxis but had no luck with the isolation of the other Proteobacteria revealed by our cultivation-independent studies. We, therefore, focussed on Sphingopyxis as this new strain was novel.

We have included the words ‘targeted isolation’ to lines 137 and 312 in order to make this information clear for readers.

  1. In the abstract, the authors indicated that most tree phyllosphere capable of using isoprene are Actinobacteria members. Are there any significant experimental changes for this study to pick up mostly Proteobacteria from the phyllosphere? Maybe highlight this experimental knowhows in Materials and Methods.  

Thank you for this comment. In previous studies, before 2018, Actinobacteria (especially Rhodococcus) were the most abundant isoprene degraders found in DNA-SIP experiments enriching with isoprene. We used the same strategies as presented in Larke-Mejia et al 2019 that used lower concentrations of isoprene (25-50 ppm) in order to isolate novel Proteobacteria. We have made the following changes to the Materials and Methods, as suggested (lines 95-99):

“DNA-SIP incubations were set up in triplicate using either 4 g soil in 40 ml of sterile water (as detailed in Larke-Mejía et al 2019 [25]) or leaf washings with 40 ml of minimal medium (as described in [27]) in 2 litres sealed glass bottles supplied with 25-50 ppmv of either 12C- or 13C-labelled isoprene, as reported in [19], in order to promote the enrichment of diverse isoprene degraders as seen in [25].”

  1. Why were the enrichment and genomic strategies focused primarily on Sphingopyxis and Gordonia? Can the isolates be named " the most abundant isoprene degraders"? 

Thank you very much for this comment. To clarify this, information can now be found in lines 317-321, as follows:

‘From the strains isolated, the latter two were of most interest because members of these families were found to be enriched during DNA-SIP experiments with leaf washings from oil palm (Figure 1). Most importantly, there was only one other isolated isoprene degrading representative from the genus Gordonia (Gordonia i37, [21]) and no Sphingopyxis have been reported previously to grow on isoprene.’

Reasons why we can’t name the isolates as “the most abundant isoprene degraders”:

  1. The isoprene enrichment was maintained for a few weeks after the DNA-SIP experiment and we do not have data that would tell us how abundant they were at the time of isolation. This would need to be the focus of a new study.
  2. Figure 2A: Sphingopyxis OPL5 was not very abundant during the time of the DNA-SIP experiment.
  3. Figure 2B: Gordonia OPL2 was not the most abundant OTU, as mentioned in line 383.

  1. Fig. 1 Consider changing the coloring patterns; it is hard to separate Methylibium from Pseudomonas and Aquincola. 

Thank you for this observation, we have now changed the colour of these three genera for clarity.

Reviewer 2 Report

The work presented is, very interesting, well written and presented and the conclusions well supported by the experiments.
Just a couple of minor  aspects that need a little clarification :

Lines 259- 261: In this sentence something seems to be missing. Please rephrase it

 Lines 310-311:  informed by the 16S rRNA ....this sentence is not understood in context .

Lines : 427 with 1 ml per liter v10 vitamins solution. is that v correct?

Please insert a figure in supplementary material including a growth curve and also the growth rate,  so that the growth between OPL5 and Gordonia OPL2 can be compared

Author Response

Lines 259- 261: In this sentence something seems to be missing. Please rephrase it

Thank you for this comment, we have changed the phrase to (Line 261-262):

“Figure 1 shows the relative abundance of bacterial 16S rRNA genes (at the genus level), obtained by PCR of enriched and un-enriched DNA extracted from oil palm soil samples and leaf washings.”

 Lines 310-311:  informed by the 16S rRNA ....this sentence is not understood in context.

Thank you for this comment, lines 312-317 have now he changed to the following:

“For the targeted isolation of isoprene degraders shown by the 16S rRNA gene profiling of bacterial communities in DNA extracted from SIP experiments (Figure 1), the DNA-SIP microcosms were continuously enriched with 25 ppmv of isoprene and samples were sub-cultured into Ewers medium supplied with 25 ppmv isoprene. Subsequent serial dilution, plating of enrichments and isoA PCR were used to confirm the presence of the isoA gene of Rhodococcus sp. strain OPL1, Gordonia sp. strain OPL2 and Sphingopyxis sp. strain OPL5 from oil palm leaf enrichments.”

Lines : 427 with 1 ml per liter v10 vitamins solution. is that v correct?

Yes, this is correct. V10 refers to the type of vitamins solution.

Please insert a figure in supplementary material including a growth curve and also the growth rate,  so that the growth between OPL5 and Gordonia OPL2 can be compared

Thank you for this comment. In line 432 we present the optimum growth rate and doubling time for Sphingopyxis OPL5. Our experiments concentrated on the growth of OPL5 and multiple growth curves and conditions were optimised for this isolate.

For Gordonia OPL2 we obtained a growth rate of 0.12 h-1 (added to line 335). This was based on growth conditions established for OPL5 (Ewers minimal medium, pH 6.5 with 1ml/l v10 vitamin solution). Detailed growth optimisation experiments would be required to establish the optimal growth rate for this Gordonia isolate and since the focus of the current manuscript was on the new Sphingopyxis strain, this would need to be the subject of a new comparative study (along with many other extant isoprene degraders).